# Elevated LSD1 and SNAIL Expression Indicate Poor Prognosis in Hypopharynx Carcinoma

**DOI:** 10.3390/ijms23095075

**Published:** 2022-05-03

**Authors:** Justus Bottner, Julika Ribbat-Idel, Luise Klapper, Tobias Jagomast, Anna-Lena Lemster, Sven Perner, Christian Idel, Jutta Kirfel

**Affiliations:** 1Institute of Pathology, University of Luebeck and University Hospital Schleswig-Holstein, Campus Luebeck, 23538 Luebeck, Germany; justus.bottner@student.uni-luebeck.de (J.B.); julika.ribbat-idel@uksh.de (J.R.-I.); luise.klapper@student.uni-luebeck.de (L.K.); tobias.jagomast@student.uni-luebeck.de (T.J.); anna-lena.lemster@uksh.de (A.-L.L.); sven.perner@uksh.de (S.P.); 2Institute of Pathology, Research Center Borstel, Leibniz Lung Center, 23845 Borstel, Germany; 3Department of Otorhinolaryngology, University of Luebeck and University Hospital Schleswig-Holstein, Campus Luebeck, 23538 Luebeck, Germany; christian.idel@uksh.de

**Keywords:** head and neck squamous cell carcinoma (HNSCC), LSD1, SNAIL, co-expression, IHC, biomarker

## Abstract

Head and neck squamous cell carcinomas (HNSCC) are among the most common cancers worldwide and are associated with a poor prognosis for patients. Among HNSCC, those originating in the hypopharynx have the worst prognosis. The histone demethylase LSD1 has been shown to promote cancer initiation, progression, and relapse through various mechanisms and is upregulated in many cancer tissues. LSD1 physically interacts with SNAIL and is required for SNAIL mediated transcriptional repression. Previous studies of the prognostic value of LSD1 in HNSCC have been limited in their analysis of sub-sites, and a correlation between LSD1 and SNAIL has not been shown in HNSCC patient samples. Here we used a large, representative, and clinically well-characterized cohort of 339 HNSCC patients to investigate the co-expression of LSD1 and SNAIL and their prognostic value in all HNSCC using immunohistochemical staining. Elevated LSD1 expression correlated with advanced tumor stage and poor progression-free survival (PFS) in HNSCC originating in the hypopharynx. Overexpression of the transcription factor SNAIL independently correlated with worse overall survival (OS) and PFS in HNSCC in general and prominently in tumors of the hypopharynx. Furthermore, increased LSD1 expression significantly correlated with elevated SNAIL expression in patient samples. Therefore, the presented data implicates LSD1 and SNAIL as independent prognostic biomarkers.

## 1. Introduction

Head and neck squamous cell carcinoma (HNSCC) is a heterogeneous group of tumors characterized by lesions in the oral cavity, larynx, pharynx, including nasopharynx, oropharynx, and hypopharynx. With an incidence of approximately 890,000 and 450,000 deaths in 2018, HNSCC is one of the most common cancers worldwide, and only about 40–50% of patients with HNSCC will survive for 5 years [1,2]. The incidence of HNSCC is estimated to increase by over 30% by 2035 [3].

Primary risk factors for their development are alcohol, tobacco consumption, environmental pollutants, and viral infections such as HPV or EBV [4]. In the early 2000s, HPV status was identified as a predictive biomarker for HNSCC of the oropharynx. As an immunohistochemical proxy for HPV infection, p16 overexpression is now used to subgroup HNSCC of the oropharynx into p16 positive and p16 negative in the TNM-classification of malignant tumors [5,6,7].

HNSCC is most commonly treated with surgery, radiation, or platinum-based chemotherapy [8]. The choice of therapy and its chances of success greatly depends on the location and stage of the tumor [9,10]. Among HNSCC, those originating in the hypopharynx have the worst prognosis and are least responsive to chemoradiation [9,11]. Despite the improvement of these therapies and the introduction of new therapeutic approaches like the use of checkpoint inhibitors, the survival of HNSCC patients has not significantly improved within the past decades [2]. Local recurrence and metastasis are limiting factors for the success of the treatment.

In general, there is a lack of prognostic and predictive biomarkers amongst HNSCC. Two-thirds of patients are initially diagnosed in advanced stages, i.e., UICC III and IV, of which only about 30% survive the first 2 years after diagnosis. Approximately half of them develop a local recurrence [2,12,13]. The development of a local recurrence is not always dependent on the UICC stage or size of the tumor at initial diagnosis, making early identification of those patients whose tumors will recur difficult.

Over the years, cancer research efforts have focused on the genetic basis of tumor development and progression, identifying mutations and characterizing pathways that activate oncogenes and inactivate tumor suppressor genes. More recently, the research has pointed to epigenetic alterations as critical changes involved in the initiation and progression of human cancers [14]. Epigenetic alterations, including DNA methylation, covalent histone modifications, chromatin remodeling, and non-coding RNAs, are frequently involved in carcinogenesis, tumor progression, and resistance to therapy.

Hypermethylation of genes such as cyclin-dependent kinase inhibitor 2A (CDKN2A) was tested as prognostic markers in HNSCC tumors. CDKN2A is involved in tumorigenesis of HNSCC tumors and encodes tumor suppressors. DNA-methylation of genes generally reduces protein expression by transcriptional silencing of the corresponding genes. However, analysis of the prognostic implication of CDKN2A methylation shows varying results [15].

Epigenetic control via histone modification is largely operated by lysine methyltransferases (KMTs) and lysine demethylases (KDMs). Lysine-specific demethylase 1 (LSD1, KDM1A) is a member of the flavin adenine dinucleotide (FAD-) dependent amine oxidase (AO) and targets mono- or di-methylated histone H3K4 and H3K9, acting as a transcriptional repressor or activator in the regulation of gene expression. Elevated levels of LSD1 show a close relationship with many cellular effects such as epithelial-mesenchymal transition (EMT), cell proliferation and differentiation, stem cell biology, and malignant transformation. LSD1 inactivation also enhances anti-tumor immunity and enables checkpoint blockade [16]. LSD1 regulates some non-histone substrates, including DNMT1, p53, STAT3, and E2F1, which play an important role in gene expression [17,18,19,20,21].

LSD1 is highly expressed in many cancer types, including breast, prostate, esophageal, bladder, and lung cancer, as well as neuroblastoma [22,23,24,25,26,27]. Within these cancers, it is often associated with advanced cancer stage and poor prognosis [22,24,26]. Pharmacological inhibition of LSD1 leads to lower proliferation, differentiation, invasion, and migration in vitro and in vivo [28].

SNAIL (SNAI1, SNAIL1) is a zinc-finger transcription factor of the SNAIL family, consisting of SNAIL, SNAIL2/SLUG, and SNAIL3. The C-terminal zinc-finger domain of the SNAIL protein functions as a DNA binding site. The N-terminal SNAG domain shows structural similarities to the N-terminus of histone H3 and functions as a binding site for a variety of co-repressors and epigenetically active protein complexes such as the histone deacetylase complex 1/2 (HDAC1/2), the polycomb repressive complex 2 (PRC2) or the LSD1/CoREST complex [29,30,31].

SNAIL has shown to be a marker for high malignancy, metastasis, and poor prognosis in various tumor entities such as breast, ovarian, endometrial, hepatocellular, bladder cancer, and some head and neck tumors [32,33,34,35,36,37,38]. Previous studies show that SNAIL leads to EMT, a process by which epithelial cells lose their polarity and are converted to a mesenchymal phenotype, and a central mechanism to induce invasiveness and metastasis of tumors. SNAIL leads to EMT via the repression of cell-adhesion molecules such as E-cadherin [39].

The interaction between LSD1 and SNAIL has been shown to be critical for the recruitment of LSD1 to the E-cadherin promoter region and the following repression of E-cadherin [40]. A correlation between LSD1 and SNAIL expression in patient samples has been demonstrated only in breast cancer tissue [29]. 

In this study, we aim to investigate the correlation of LSD1 and SNAIL expression levels in HNSCC, their association with tumor and patient characteristics, as well as their potential prognostic value for HNSCC patients by examination of the expression levels using immunohistochemical staining. 

## 2. Results

### 2.1. Expression of LSD1 and SNAIL 

The LSD1 and SNAIL expression was analyzed in the tissue of primary tumors (*n* = 339), local recurrences (*n* = 61), lymph node metastases (*n* = 165), as well as distant metastases (*n* = 22) (Appendix A). LSD1 and SNAIL were expressed in tumor cells but not expressed in nonmalignant cells, such as stromal tissue.

The stained cohort showed a broad spectrum of staining intensities ranging from completely negative to strongly positive for the LSD1 as well as the SNAIL staining (Figure 1). Most cores were either completely negative or completely positive, ranging from weakly positive to moderately positive and strongly positive staining intensities. For the LSD1 staining, the variance of staining intensity within each core and the three cores belonging to the same sample was low, with most nuclei falling into the same category of intensity (intracase homogeneity). The variance of SNAIL-staining intensity within individual TMA cores was noticeably higher compared to the LSD1 staining (relative intracase heterogeneity). Most cores contained weakly, moderately, and highly positively stained nuclei, with the relative percentages varying between respective samples. Hence, the H-Score was used to assess the average staining intensity.

### 2.2. LSD1 and SNAIL Expression Differ between Tissue Types

The average LSD1 expression of primary tumors was significantly lower than that of local recurrences and higher than that of lymph node metastases (*p* = 2.1 × 10^−4^ and *p* = 0.022, respectively) (Figure 2a). These correlations remained significant when the LSD1 expression of primary tumors was compared to that of the corresponding local recurrence and lymph node metastasis (*p* = 7.3 × 10^−4^ and *p* = 0.020, respectively). There were no significant differences between the average LSD1 expression of primary tumors and distant metastases or the matched-pair analysis of primary tumors and the correlating distant metastasis.

The average SNAIL expression was shown to be significantly lower in primary tumors than in metastases, as well as local recurrences (LN: *p* = 0.034, DM: *p* = 0.025, LR: *p* = 3.0 × 10^−9^) (Figure 2c). In a matched-pair analysis of primary tumors and their corresponding metastases and recurrences, only the differences in SNAIL expression between primaries and metastases were shown to remain significant (LN: *p* = 0.034 DM: *p* = 0.019), whereas the difference between primaries and recurrences was not significant in matched-pair analysis (*p* = 0.405).

### 2.3. Higher LSD1 Expression in Hypopharyngeal HNSCC

We further examined if there were differences in the LSD1 expression levels between different sites of origin in primary HNSCC using tissue samples from primary tumors (*n* = 339) originating from the hypopharynx (*n* = 46), larynx (*n* = 98), oral cavity (*n* = 77), and oropharynx (*n* = 107, *n*(*p16+*) =56, *n*(*p16−*) =51). Since p16 positive and p16 negative tumors of the oropharynx did not show a significant difference in LSD1 or SNAIL expression (*p* = 0.55 and *p* = 0.26, respectively), they were grouped for this analysis (Appendix A).

Primary tumors originating in the hypopharynx showed the highest LSD1 expression levels, which were significantly higher compared to those of primary tumors originating in the larynx, oral cavity, or oropharynx (*p* = 0.046, *p* = 0.008, *p* = 0.049, respectively) (Figure 2b). The LSD1 expression levels in laryngeal, oral, and oropharyngeal primary HNSCC were nearly equal and showed no significant differences.

In contrast, the SNAIL expression level was not significantly higher in any of the primary tumor sites (Figure 2d).

### 2.4. Correlation of LSD1 and SNAIL with Clinical Data and Established Risk Factors 

Analysis of average LSD1 expression in HNSCC tumors categorized by various tumor characteristics and risk factors revealed a correlation of LSD1 expression with advanced tumor stage (Table 1). 

LSD1 expression is significantly higher in advanced T, N, and UICC stages than in lower stages (*p* = 3.8 × 10^−4^, *p* = 0.008, *p* = 0.018, respectively). Furthermore, LSD1 expression is higher in tumors of patients who consume alcohol than in those who do not (*p* = 0.015).

When HNSCC tumors are considered separately according to their primary site, the correlation of LSD1 expression with advanced tumor stage can be observed exclusively in tumors of the hypopharynx (T stage: *p* = 9.5 × 10^−4^, UICC: *p* = 0.020), whereas no significant differences dependent on tumor stage (T, UICC) were observed in tumors of the other primary sites. 

The correlation of LSD1 expression with the N stage that was shown for HNSCC in general, with higher N stages having higher LSD1 expression levels, could also be seen in SCC of the hypopharynx and p16 positive SCC of the oropharynx when analyzed separately (*p* = 0.021 and *p* = 0.006, respectively). 

In SCC of the larynx, increased LSD1 expression was seen in tumors from female patients and those without nicotine abuse (*p* = 0.009 and *p* = 0.027, respectively).

In addition, in oral SCC, higher LSD1 expression could be observed in tumors from patients consuming alcohol than in those not consuming alcohol (*p* = 0.002).

The SNAIL expression analysis concerning the different tumor and patient characteristics showed a significant inverse correlation of SNAIL expression with the N stage as well as the nicotine consumption of the patients when considering the entire cohort (*p* = 0.018 and *p* = 0.024, respectively). Here, primary tumors from patients without lymph node metastases and non-smokers had higher SNAIL expression (Table 2). 

A correlation of SNAIL expression with N0 stage was not seen in the isolated analysis of the different sites of origin. Although not statistically significant, oropharyngeal, laryngeal, and oral SCC showed the same tendency as the cohort as a whole, whereas the tendency was reversed in tumors of hypopharyngeal origin.

In laryngeal SCC, SNAIL expression was higher in female patients, p16 positive and less advanced (UICC I or II) tumors as well as those from patients who are non-smokers (*p* = 0.043, *p* = 0.011, *p* = 0.034, and *p* = 0.016, respectively). 

Furthermore, correlation analysis of LSD1 and SNAIL revealed their co-expression in primary HNSCC (PCC = 0.3237, *p* = 2.067 × 10^−8^) (Table 3) as well as across all available samples (PCC = 0.2982, *p* = 3.611 × 10^−13^) (Table 4). This finding was in line with the perceived impression that cores that showed intense LSD1 staining tended to also show high SNAIL staining intensity (Figure 3).

### 2.5. High LSD1 and SNAIL Expression Indicate Poor Prognosis

We investigated whether elevated LSD1 or SNAIL expression correlated with overall survival (OS) or progression-free survival (PFS) in HNSCC patients in general or any of the primary tumor sites when analyzed individually.

There were no significant differences regarding patient survival between primary tumors with low and high LSD1 expression in HNSCC in general. (Figure 4). However, elevated LSD1 expression correlated with shorter PFS in hypopharyngeal HNSCC patients (Figure 5b) and is independent of other prognostic tumor characteristics such as T, N, or UICC stage (Appendix A).

There were no correlations between LSD1 expression and survival in tumors of the larynx, oral cavity, or the oropharynx (data not shown).

Increased SNAIL expression in primary tumors correlated with worse prognosis in HNSCC regarding the OS as well as the PFS (Figure 4). This prognostic quality is independent of prognostic tumor characteristics (Appendix A). When analyzing the primary tumor sites separately, high SNAIL expression correlated with shorter OS as well as PFS only in HNSCC of the hypopharynx (Figure 5).

In addition, we evaluated the prognostic relevance of a combination of high LSD1 and high SNAIL expression.

HNSCC patients with tumors showing both high LSD1 and high SNAIL expression correlated with a worse prognosis regarding the PFS in HNSCC in general, while there was no significant difference regarding the OS (Figure 4). 

Looking at the primary tumor sites separately, high LSD1 and SNAIL expressing HNSCC had a shorter OS and PFS only in patients with hypopharyngeal (Figure 5) but not in those with laryngeal, oral, or oropharyngeal HNSCC (data not shown).

When analyzing the survival for each primary tumor site individually, there was no significant difference between low and high LSD1, SNAIL, or LSD1 and SNAIL expression in patients with tumors originating in the larynx, oral cavity, or oropharynx.

However, in patients with primary hypopharyngeal tumors, high expression levels of SNAIL or both (LSD1 and SNAIL) correlated with shorter OS (Figure 5b). Elevated expression levels of either LSD1, SNAIL, or both LSD1 and SNAIL also correlated with shorter PFS.

## 3. Discussion

Here, we have analyzed the prognostic and predictive qualities of LSD1 and SNAIL, their correlation with clinical data, and, for the first time, investigated the co-expression of LSD1 and SNAIL in HNSCC. The interaction of SNAIL and LSD1, in which the SNAG domain of the SNAIL protein functions as a molecular hook to recruit LSD1 to its target gene, where it represses gene expression and induces EMT, has previously been demonstrated only in breast cancer [29].

Correlations of LSD1 and SNAIL expression with tumor and patient characteristics as well as patient survival showed site-specific differences regarding both SNAIL and LSD1 expression. The increased LSD1 and SNAIL expression in female patients, solely observed in laryngeal SCC, may be due to increased levels of the female sex hormone estrogen. The larynx is an organ that is strongly influenced by sex hormones during puberty. Several studies have shown the influence of estrogen on tumorigenesis in various carcinomas such as breast, endometrial, and colorectal cancer [41,42,43]. In endometrial carcinomas, estrogen-induced LSD1 expression levels via G-protein-coupled estrogen receptor 1 (GPER) [44]. The GPER is also expressed in SCC of the larynx, where its inhibition leads to inhibition of proliferation and migration [45].

Furthermore, the correlation of SNAIL expression with lower tumor stages (UICC I and II) in laryngeal carcinomas may be related to the expression of estrogen receptors alpha (ERα) and beta (ERβ). In ovarian carcinomas, SNAIL expression is increased by ERα activation and decreased by activation of Erβ [46]. The ratio of ERβ to ERα expression is increased in laryngeal SCC compared to oral and pharyngeal SCC, and ERβ expression has been shown to correlate with a low TNM stage and increased levels of E-cadherin in SCC of the larynx [47,48].

Contrary to previous studies showing an increase of EMT and SNAIL expression due to nicotine exposure in HNSCC cells and healthy oral epithelial cells, we showed in laryngeal SCC of nonsmoking patients a higher SNAIL as well as LSD1 expression compared to those of smokers [49,50]. This association is limited to laryngeal SCC and requires further evaluation to explore the localization-dependent influence of nicotine on tumorigenesis and characteristics.

Alcohol is a known risk factor for the development of HNSCC of the pharynx and oral cavity, with high alcohol consumption being associated with an increased relative risk of carcinogenesis compared to no or moderate alcohol consumption [51]. In our study, we showed increased LSD1 expression in the oral tumors of patients who consumed alcohol than in those who did not. A correlation of LSD1 expression with tumor stage or survival in oral SCC, as shown by Wang et al., could not be seen in our cohort [52]. 

Wakisaka et al. demonstrated increased EMT in HPV positive oropharyngeal SCC with advanced N stage [53]. This observation is consistent with the increased LSD1 expression in p16 positive oropharyngeal SCC with advanced N stage shown in our analysis.

Our data demonstrated higher SNAIL expression in N0 stage tumors, while LSD1 correlated with advanced N stage. This trend can be seen particularly in SCC of the oropharynx. Based on their association with EMT and tumor cell migration, we hypothesized SNAIL as well as LSD1 expression to correlate with advanced N stage. Further analysis of the interplay of LSD1 and SNAIL during the development of metastases is required to understand their function in this process. 

Interestingly, the average LSD1 expression was significantly higher in hypopharyngeal SCC compared to other primary localizations. In addition, we investigated a significant correlation of LSD1 expression with advanced tumor stage (T and UICC stage) in HNSCC in general and also in hypopharyngeal SCC. Furthermore, only hypopharyngeal tumors showed a correlation of LSD1 with poor PFS. High expression levels of SNAIL showed a correlation with poor OS and PFS in HNSCC, in general, that was independent of other prognostic tumor and patient characteristics, e.g., T and UICC stage or the p16 status. A correlation of elevated SNAIL expression with poor prognosis was only seen in HNSCC of the hypopharynx. Further, combined overexpression of LSD1 and SNAIL correlates with poor prognosis regarding the OS as well as the PFS in HNSCC in general and in hypopharyngeal HNSCC. The predictive value of LSD1 and SNAIL co-expression seems to reflect only the predictive value of SNAIL when considering HNSCC as a whole but appears to have an additive effect when considered in SCC of the hypopharynx.

Despite the roundup of HNSCC as one cancer entity, recent data indicate that they might actually be a rather heterogeneous group of tumors. Risk factors for the development of the tumors differ in a localization-specific manner, i.e., toxic versus viral carcinogenesis in the oral cavity and oropharynx HNSCC, respectively [10]. In a previous study, we showed that EVI1 is differentially expressed in hypopharynx versus non-hypopharynx HNSCC [54]. Prognosis, incidence, and therapy sensitivity also differ depending on the location of the primary tumor and further subclassification based on HPV status and molecular subtypes [55,56,57]. The discovery of the prognostic value of HPV status amongst HNSCC of the oropharynx led to the subgrouping of oropharyngeal SCC into HPV positive and HPV negative [5]. There is still, however, a lack of prognostic and predictive biomarkers for other HNSCC sub-sites. 

Previous analyses of LSD1, as well as SNAIL expression in HNSCC, have shown them to be linked to advanced tumor stage and worse prognosis but were limited to oral SCC or lacked the separate consideration of different HNSCC subsites [38,52,58]. Our data showed a localization-dependent prognostic relevance of LSD1 and SNAIL expression levels as well as their co-expression and suggested a separate consideration of hypopharyngeal SCC as its own entity. LSD1 and SNAIL as well as their co-expression, can be used as a marker for poor prognosis and tumor progression in HNSCC of the hypopharynx, and screening intervals of tumors with high expression could be adjusted accordingly.

Recently, it was demonstrated that SNAIL plays an important role in radio- and cisplatin-based chemotherapy resistance as well as cancer cell stemness in HNSCC [59,60]. LSD1 has been shown to promote cancer cell stemness via different and partially tissue-specific mechanisms in several cancers, such as breast cancer, hepatocellular carcinoma, and colorectal cancer, as well as HNSCC [58,61,62,63]. Cancer stem cells are considered to have increased resistance to conventional treatments such as radiation and chemotherapy, which is why they are thought to promote cancer recurrence and metastasis [64]. LSD1 inhibition in vitro has also been shown to increase chemosensitivity in several solid tumors, including oral SCC [52,65,66,67]. Furthermore, Han et al. recently reported that HNSCC with high LSD1 expression has a worse prognosis. Inhibition of LSD1 is a promising strategy to sensitize HNSCC to PD-1 blockade by suppressing stem-cell-like properties [58].

Since especially advanced HNSCC originating in the hypopharynx show poor response to treatment with conventional radio-chemotherapy, further analysis of a correlation between LSD1 and SNAIL expression and resistance to these treatment options in vivo could potentially lead to a personalization of therapeutic approaches [9]. 

In addition to the prognostic value of LSD1 and SNAIL, we were able to show a highly significant correlation between LSD1 and SNAIL in HNSCC patient samples. This correlation was seen when all available tissue samples were considered and in the separate consideration of primary tumors. The importance of the interaction of LSD1 and SNAIL for their role in EMT and, therefore, tumor progression was previously shown in breast cancer by Lin et al. [40]. The correlation of LSD1 and SNAIL expression suggests that LSD 1 and SNAIL are associated in patient tissue and imply that they are meaningfully linked in vivo.

LSD1 and SNAIL are generally associated with repression of E-cadherin, which is highly expressed in the metastases of tumors compared to the respective primaries [68,69]. In accordance, we investigated that LSD1 was downregulated in lymph node metastases compared to the corresponding primary tumor. Contrary to our assumption that SNAIL expression might also be downregulated in metastatic tissue in order to induce E-cadherin expression and thereby facilitate the shift from a mesenchymal to an epithelial phenotype during metastasis formation, SNAIL expression was significantly higher in lymph node- and distant metastases compared to corresponding primary tumors. The role of E-cadherin and its necessity, as well as the concept of mesenchymal-epithelial transition as a mechanism for metastatic colonization, is not fully understood and currently debated [70,71]. Elevated SNAIL expression in lymph node metastases in relation to corresponding primaries, while the expression of EMT markers showed no significant difference, was also described in breast cancer [72]. Here, the authors proposed the elevated levels of SNAIL to possibly be part of a mechanism that facilitates cancer cell survival at the metastatic site by decreasing the cell division rate. Further analysis is needed to understand the interaction of cancer cells with the metastatic site and the involvement of SNAIL and its interaction with LSD1 in the development of metastases.

Recently, Jorgensen et al. described a shift from an epithelial to a mesenchymal (or intermediate) phenotype from primary tumors to recurrences based on the expression levels of EMT markers such as E-cadherin, vimentin, Twist, and N-cadherin in breast cancer [73]. This was attributed to tumor progression and is in line with our finding that LSD1 expression was higher in local recurrences compared to matched primary tumors and a correlation of LSD1 with advanced tumor stages.

In summary, we provide the first evidence that LSD1 and SNAIL are overexpressed in SCC of the hypopharynx, associated with poor prognosis, and may serve as a novel drug target in this tumor entity.

## 4. Materials and Methods

### 4.1. Cohort and TMA

We used the vast and well-characterized HNSCC cohort that was previously established by our group [54,74,75].

All patients were diagnosed between 2012 and 2015 in our Institute of Pathology and treated in the on-campus Department for Otorhinolaryngology of the University Hospital Schleswig-Holstein in Luebeck. The study was conducted according to the guidelines of the Declaration of Helsinki and approved by the Ethics Committee of the University of Luebeck (protocol code 16-277).

Clinical data of 339 patients were retrospectively obtained from medical records and afterward anonymized. Tumor stages at the time of tumor diagnosis were re-assessed according to the 8th edition TNM classification for HNSCC. The detailed composition of the cohort is described in Appendix A. 

Formalin-fixed paraffin-embedded (FFPE) tumor tissues were retrieved from the archives of the Institute of Pathology of the University Hospital Schleswig-Holstein, Campus Luebeck. Tissue samples of primary tumors, lymph nodes, and distant metastases, as well as local recurrences, were arranged to create tissue microarrays (TMAs). Using a semiautomatic tissue arrayer (Beecher Instruments, Sun Prairie, WI, USA), representative cores from donor blocks were placed into TMA recipient blocks, as previously described [76,77]. Each TMA carried up to 54 tumor samples with up to 6 normal tissue samples collected from non-tumorous areas of tissue samples. Each sample was a triplet of cores with an area of 1 mm^2^.

### 4.2. Immunohistochemical Staining and Evaluation

Immunohistochemical staining was conducted using the Ventana benchmark ultra automated staining system using Ventana reagents and the Ventana OptiView DAB Detection Kit (v6) (Roche, Basel, Switzerland). Heat-mediated antigen retrieval was performed for 32 min at 92 °C for both LSD1 and SNAIL staining using Cell Conditioning Solution 1 (Ventana CC1; #950-124) for LSD1 and Cell Conditioning Solution 2 (Ventana CC2; #950-123) for SNAIL staining. LSD1 antibody 1B2E5 (monoclonal, mouse, dilution 1:350, nb 100 1762; Novus Biological, Littleton, CO, USA) and the SNAIL antibody AF6032 (polyclonal, rabbit, dilution 1:1000, AF6032; Affinity Biosciences, Cincinnati, OH, USA) were used as primary antibodies.

For scanning and digitalizing the LSD1-stained TMA slides, the Ventana iScan HT scanner (Roche, Basel, Switzerland) was used. The SNAIL-stained TMAs were scanned using the Ventana DP 200 slide scanner (Roche, Basel, Switzerland). The TMAs were then analyzed using the semi-automated bioimaging software QuPath (v0.2.3) [78].

To analyze only the staining intensity of tumor cells and not the surrounding benign tissue or stromal cells, tumor cell areas were manually annotated as regions of interest (ROI). These annotations were either performed or verified by a board-certified pathologist.

Using the positive cell detection tool in QuPath, cells and their nuclei were detected. The staining intensity of each nucleus was then categorized as either negative, weak, moderate, or strong positive according to manually set thresholds. For each sample consisting of 3 cores, an H-score ranging from 0 to 300 was calculated by adding the percentage of highly stained nuclei multiplied by 3, the percentage of moderately stained nuclei multiplied by 2, and the percentage of weakly stained nuclei multiplied by 1.
H−Score=3×strong pos. %+2×mod. pos. %+1×weak pos.%

### 4.3. Statistical Analysis 

For statistical analysis, we used the statistics interface jamovi (v1.6, The jamovi Project, Sydney, Australia, https://www.jamovi.org, accesed on 9 August 2021), which is built on top of the R statistical language (R Core Team (2020). *R: A Language and environment for statistical computing*. (Version 4.0), https://cran.r-project.org, accesed on 9 August 2021). The following packages were used: *Survival* (Terry M Therneau (2020). *A Package for Survival Analysis in R.*, https://cran.r-project.org/package=survival, accesed on 9 August 2021), *finalfit* (https://cran.r-project.org/package=finalfit, accesed on 9 August 2021).

For the categorization of low and high LSD1 and SNAIL expression, an H-score above the respective median, rounded to four decimal places, was considered high, and an H-score below that median was considered low.

We applied Mann–Whitney tests to compare the LSD1 and SNAIL expression in different tissue types and primary locations, different clinicopathological features, and risk factors. For the correlation of LSD1 and SNAIL measurements, the Pearson correlation coefficient (PCC) was calculated, and the association of categorized high/low LSD1 and SNAIL expression was tested using Pearson´s Chi-squared test. Wilcoxon signed-rank test was applied to compare the LSD1 expression levels between primary tumors and their matched recurrence/metastases. The Kaplan–Meier method and log-rank testing were used to calculate 60-months OS and PFS and to test for statistical significance. Univariate and multivariate survival analysis was performed using the Cox proportional hazards regression model, whereas only tumor characteristics and risk factors showing significance in the univariate analysis were included in the multivariate survival analyses. All required assumptions were checked prior to statistical testing. 

A *p*-value below or equal to 0.05 was considered statistically significant and *p*-values marked as follows: *-*p* < 0.05, **-*p* < 0.01, ***-*p* < 0.001, ****-*p* < 0.0001. 

## Figures and Tables

**Figure 1 ijms-23-05075-f001:**
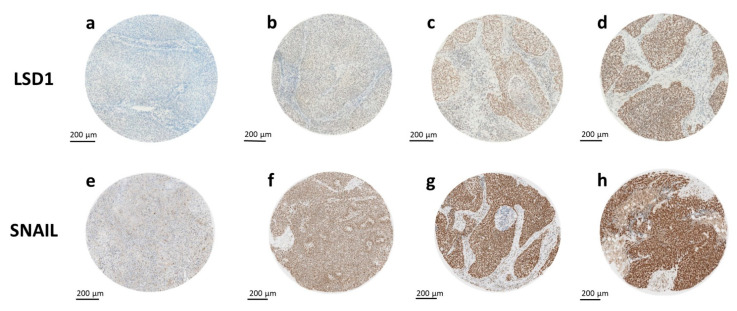
Staining patterns for LSD1 and SNAIL. LSD1 and SNAIL expressions were assessed using immunohistochemical staining. The staining intensities varied from (**a**,**e**) negative to (**b**,**f**) weak, (**c**,**g**) moderate, and (**d**,**h**) high positive nuclear staining for (**a**–**d**) LSD1 and (**e**–**h**) SNAIL.

**Figure 2 ijms-23-05075-f002:**
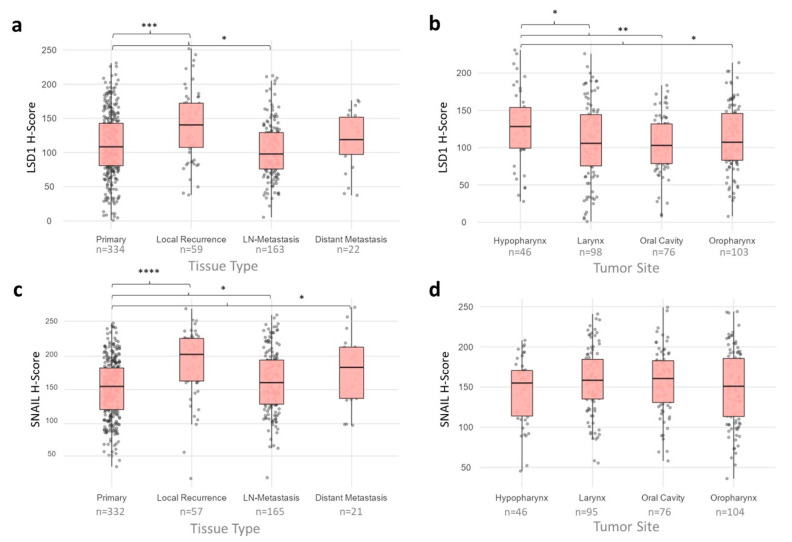
LSD1 and SNAIL expression by tissue type and by primary tumor site. LSD1 and SNAIL expression were assessed in tissue types and primary tumor sites using the H-Score. (**a**) Primary tumors show a lower LSD1 expression compared to local recurrences and higher expression compared to LN-metastases. (**b**) Primary tumors that originate in the hypopharynx show a significantly higher LSD1 expression compared to laryngeal, oral, or oropharyngeal head and neck squamous cell carcinoma (HNSCC). (**c**) Primary tumors have a significantly lower SNAIL expression than local recurrences as well as LN- and distant metastases. (**d**) There were no significant differences in SNAIL expression between primary tumor locations. *p*-values were calculated using Mann–Whitney tests and are indicated: *-*p* < 0.05, **-*p* < 0.01, ***-*p* < 0.001, ****-*p* < 0.0001.

**Figure 3 ijms-23-05075-f003:**
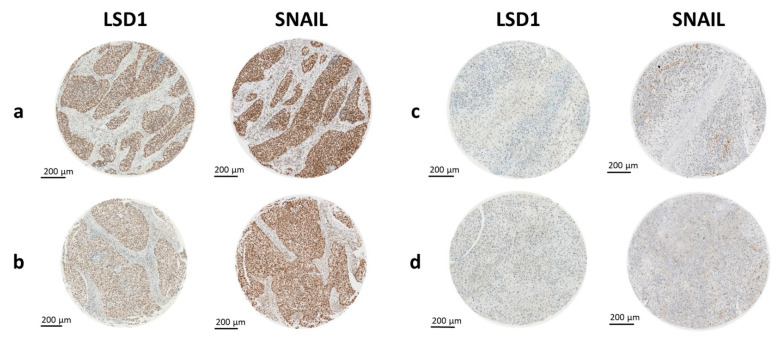
LSD1 and SNAIL staining in matched cores. LSD1 and SNAIL stained cores of identical samples showed similar staining intensities. (**a**,**b**) Samples were considered negative for both LSD1 and SNAIL. (**c**,**d**) Samples considered moderate or high positive for both LSD1 and SNAIL.

**Figure 4 ijms-23-05075-f004:**
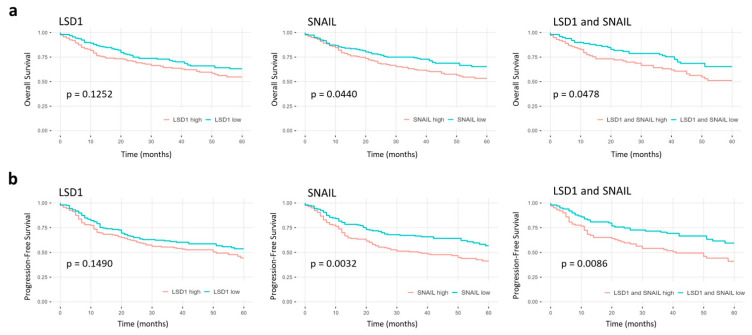
Kaplan–Meier analysis of primary HNSCC in general. Five-year survival curves showing the differences in (**a**) overall and (**b**) progression-free survival between low and high LSD1, SNAIL, or combined LSD1 and SNAIL expression in primary HNSCC. The corresponding *p*-values are shown in each panel and were calculated using two-sided log-rank testing.

**Figure 5 ijms-23-05075-f005:**
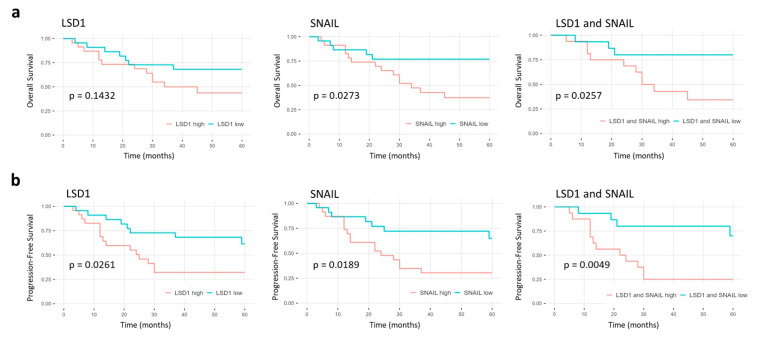
Kaplan–Meier survival analysis of primary hypopharyngeal HNSCC. Primary tumors originating in the hypopharynx were categorized into groups showing low and high LSD1, SNAIL, or combined LSD1 and SNAIL expression, and their differences in (**a**) overall and (**b**) progression-free survival were assessed using the Kaplan–Meier method. The corresponding *p*-values are shown in each panel and were calculated using two-sided log-rank testing.

**Table 1 ijms-23-05075-t001:** LSD1 expression in primary HNSCC tumors by tumor site and patient and tumor characteristics. Shown are the average H-Score values of primary HNSCC separated by the site of origin and categorized by patient and tumor characteristics. Respective H-Score values of categories showing significant differences are bold. *p*-values were calculated using Mann–Whitney tests and are indicated: *-*p* < 0.05, **-*p* < 0.01, ***-*p* < 0.001.

		HNSCC	Hypopharynx	Larynx	Oral Cavity	Oropharynx p16 pos	Oropharynx p16 neg
**Sex**	**male**	112.4	129.6	**104.2 ****	110.4	114.4	126.8
	**female**	110.2	119.5	**144.5 ****	94.9	98.6	111.6
**Age**	**≤61y.**	110.5	124.6	102.4	105.9	108.2	123.8
	**>61y.**	112.8	130.5	113.1	103.2	110.1	107.3
**T-Stage**	**T 1,2,3**	**106.8 *****	**114.3 *****	104.9	103.7	106.8	111.1
	**T 4**	**130.2 *****	**171.7 *****	119.3	111.2	123.5	128.4
**N-Stage**	**N 0,1**	**105.5 ****	**107.3 ***	109.2	104.0	**97.6 ****	109.3
	**N 2,3**	**120.0 ****	**141.9 ***	105.5	106.0	**132.1 ****	115.5
**M-Stage**	**M0**	111.7	126.7	108.7	104.1	109.3	114.7
	**M1**	120.0	188.5	109.3	113.5	-	123.7
**UICC**	**I–III**	**106.6 ***	**106.1 ***	111.8	102.5	103.5	113.3
	**IV**	**118.3 ***	**142.1 ***	104.5	107.9	134.4	115.8
**p16**	**pos**	113.3	98.5	126.9	114.1	109.3	-
	**neg**	111.4	130.8	105.5	103.3	-	114.9
**Alcohol**	**Yes**	**119.8 ***	133.0	108.9	**120.0 ****	124.8	119.8
	**No**	**106.7 ***	117.5	108.7	**92.8 ****	107.2	106.2
**Nicotine**	**Yes**	113.0	128.2	**107.5 ***	105.5	114.0	116.2
	**No**	115.7	141.7	**145.4 ***	107.7	100.7	92.8

**Table 2 ijms-23-05075-t002:** SNAIL expression in primary HNSCC by the site of origin and patient and tumor characteristics. Shown are the average H-Score values of primary HNSCC in general and all primary tumor locations separately by tumor and patient characteristics. Categories showing significant differences are bold. *p*-values were calculated using Mann–Whitney tests and are indicated: *-*p* < 0.05.

		HNSCC	Hypopharynx	Larynx	Oral Cavity	Oropharynx p16 pos	Oropharynx p16 neg
**Sex**	**male**	151.1	147.2	**154.5 ***	157.0	145.0	150.4
	**female**	157.4	130.5	**179.3 ***	157.3	149.3	166.8
**Age**	**≤61y.**	151.6	143.9	154.0	154.8	149.6	157.6
	**>61y.**	153.0	145.2	159.8	160.0	144.1	150.4
**T-Stage**	**T 1,2**	156.3	144.6	165.0	160.2	150.5	156.6
	**T 3,4**	148.4	144.8	151.4	154.0	137.5	150.1
**N-Stage**	**N 0**	**159.5 ***	138.6	159.0	161.3	163.4	171.1
	**N 1,2,3**	**146.6 ***	145.9	153.5	152.9	142.2	146.8
**M-Stage**	**M0**	152.3	143.6	157.6	158.7	146.3	153.7
	**M1**	148.8	192.9	154.6	134.5	-	162.1
**UICC**	**I–II**	157.0	140.9	**170.3 ***	165.0	147.5	157.0
	**III–IV**	149.2	145.6	**150.8 ***	153.0	142.5	152.7
**p16**	**pos**	150.2	120.8	**181.6 ***	141.8	146.3	-
	**neg**	153.2	147.0	**152.9 ***	159.4	-	153.9
**Alcohol**	**Yes**	152.2	148.8	160.8	153.1	155.2	153.6
	**No**	151.6	133.2	156.0	159.2	144.2	154.4
**Nicotine**	**Yes**	**150.1 ***	139.9	**154.2 ***	154.3	146.2	155.7
	**No**	**167.4 ***	184.8	**188.2 ***	168.2	150.5	153.2

**Table 3 ijms-23-05075-t003:** Association between LSD1 and SNAIL expression in primary HNSCC. High LSD1 expression is significantly associated with high SNAIL expression in primary HNSCC. The *p*-value was calculated using Pearsons’s Chi-squared test.

	SNAIL Low (*n* = 164)	SNAIL High (*n* = 163)	
**LSD1 low (*n* = 163)**	104.0 (63.4%)	59.0 (36.2%)	*p* < 0.001
**LSD1 high (*n* = 164)**	60.0 (36.6%)	104.0 (63.8%)	

**Table 4 ijms-23-05075-t004:** Association between LSD1 and SNAIL expression considering all available tissue samples. LSD1 and SNAIL expression are significantly associated when all available tissue samples are considered. The *p*-value was calculated using Pearsons’s Chi-squared test.

	SNAIL Low (*n* = 285)	SNAIL High (*n* = 285)	
**LSD1 low (*n* = 288)**	177.0 (62.1%)	111.0 (38.9%)	*p* < 0.001
**LSD1 high (*n* = 282)**	108.0 (37.9%)	174.0 (61.1%)	

## Data Availability

All datasets generated and analyzed during the current study are available from the corresponding author on request.

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
