# Peer review of "Elevated LSD1 and SNAIL Expression Indicate Poor Prognosis in Hypopharynx Carcinoma"

_ijms, 2022, doi:10.3390/ijms23095075_

Round 1

Reviewer 1 Report

The manuscript “Elevated LSD1 and SNAIL Expression Indicate Poor Prognosis in Hypopharynx Carcinoma” by Bottner et al. presents data on correlation of expression of LSD1 and SNAIL markers in HNSCC patients. To perform this analysis the authors used the representative HNSCC cohort of 339 HNSCC patients. They found that elevated LSD1 expression correlated with advanced tumor stage and poor progression-free survival in HNSCC originating in the hypopharynx. Similarly, overexpression of SNAIL independently correlated with worse overall survival and poor progression-free survival in HNSCC in general and prominently in tumors of the hypopharynx. Furthermore, they showed that increased expression of LSD1 correlated with elevated SNAIL expression. Based on these findings they proposed LSD1 and SNAIL as novel therapeutic targets in HNSCC patients.

Overall, the manuscript is well written and can be published in the present form.

Author Response

We thank the Reviewer for the positive evaluation on our article.

Reviewer 2 Report

ijms-1713730
Justus Bottner et al. report that LSD1 and SNAIL are independent prognostic biomarkers for HNSCC. I think this is an interesting study and the manuscript is well-written. However, following points should be addressed to strength the findings.
1)    Histopathological finings regarding differentiation of HNSCC and presence/absence of EMT should be mentioned if available.
2)    Photo(s) of hematoxylina and eosin-stained HNSCC can be added.

Author Response

We appreciate the Reviewer’s detailed and positive evaluation on our article. In the revised manuscript, his / her specific comments have been addressed as follows:

  • English language and style are fine/minor spell check required.

This is a good point. We employed a professional English spell check (Grammarly Inc.) to address language and grammar issues. Revisions made to the manuscript have marked up using the

“Track Changes” function.

  • Histopathological finings regarding differentiation of HNSCC and presence/absence of EMT should be mentioned if available.

We agree that this point is an interesting issue that deserves consideration. EMT is not a fixed and irreversible processbut instead can be context-dependent, occurring indistinct cellular populations at particular sites within the tumour. Reversal of EMT may be necessary for the establishment of macrometastatic tumour sites, underscoring the plasticity of the process. Furthermore, tumour cells may exhibit only partial EMT, a partial acquisition of EMT markers properties.

One of the hallmarks of EMT is the functional loss of E-cadherin, which is thought to be a metastatic suppressor during tumor progression. Snail is a prominent inducer of EMT and strongly represses E-cadherin expression. Until today, we investigate only Snail as EMT marker, but not the expression of E-cadherin or N-cadherin. Therefore, we cannot provide these informations.

  • Photo(s) of hematoxylina and eosin-stained HNSCC can be added.

We added as supplemetary Material Figure S1 displaying examplary HE-stained TMA cores.